# BVS: Bayesian Visual Search with Multimodal Large Language Model for Fine-grained Perception

Geng Li [1]   Yuxin Peng [1]

## Abstract

While Multimodal Large Language Models (MLLMs) demonstrate impressive general capabilities, they struggle with fine-grained perception in ultra-high-resolution (UHR) images, particularly for tiny objects in cluttered scenes. Existing methods face a dilemma: they either rely on inefficient prior-free scanning, or depend on static prior-driven heuristics that lack posterior correction to rectify initial model biases. To address this, we propose **BVS** (**B**ayesian **V**isual **S**earch), a framework that formulates perception as a global optimization problem over a continuous spatial-scale manifold. Specifically, BVS bridges prior guidance with posterior correction: it utilizes an early-stop attention rollout of MLLM to construct reasoning-aware priors, while employing a scale-aware non-stationary kernel and GP-UCB to dynamically rectify noise and recover missing information in the prior through iterative local observations. We provide theoretical guarantees via sublinear regret bounds, and extensive experiments demonstrate that BVS significantly outperforms state-of-the-art baselines with a superior trade-off between accuracy and efficiency.

## 1. Introduction

Multimodal Large Language Models (MLLMs) have achieved significant progress in tasks such as visual question answering (Antol et al., 2015), image captioning (Li et al., 2023), and visual grounding (Lai et al., 2024). With the increase in training data and architectural optimizations (Tian et al., 2023), MLLMs demonstrate increasingly strong capabilities in visual perception (Wang et al., 2024b; Chen et al., 2024b; Bai et al., 2025a; Wang et al., 2025c; Peng et al., 2025; Xiao et al., 2026). However, despite these advance-

ments, fine-grained perception tasks in ultra-high-resolution (UHR) and cluttered dense environments (Wu & Xie, 2024; Wang et al., 2025b; Lai et al., 2025) remain a challenge for state-of-the-art MLLMs (Bai et al., 2025a; Wang et al., 2025c). These tasks typically require identifying single or multiple tiny objects (occupying merely 0.2% to 1% of the image area) within image exceeding millions of pixels, and then inferring their attributes or spatial relationships. Although existing MLLMs generally support dynamic resolution (Bai et al., 2025a;b; Wang et al., 2024a; Zhu et al., 2025; Wang et al., 2025c), the massive number of visual tokens generated by high-resolution images continues to limit model performance.

To address this bottleneck, various visual search strategies have been developed to identify query-relevant sub-regions. These methods allow models to focus on critical visual details while filtering out irrelevant background information. According to utility of prior distribution, they can be categorized into two main groups: *Prior-free Visual Search Methods* and *Prior-driven Visual Search Methods*.

*Prior-free Visual Search Methods* do not rely on explicit assumptions regarding the distribution of semantically salient regions. Instead, they typically employ traditional search algorithms, such as Best-First Search, Monte Carlo Tree Search (MCTS), or greedy search to explore visual regions from scratch with an evaluation function guided. SEAL (Wu & Xie, 2024) pioneered this direction by integrating a trainable scoring head into LLaVA (Liu et al., 2023) to perform a greedy hierarchical grid search. To accelerate this process, subsequent approaches adopted training-free evaluation metrics and structured high-resolution images into hierarchical trees, utilizing algorithms like Best-First Search (Shen et al., 2024) or MCTS (Li et al., 2025) for efficient traversal. Alternatively, a series of end-to-end grounding and reasoning methods, such as DeepEyes (Zheng et al., 2025), Thyme (Zhang et al., 2025b), mini-o3 (Lai et al., 2025), and PixelReasoner (Su et al., 2025) are proposed. They learn search policies directly from data. And during inference, they employ a greedy strategy to select sub-images based on the grounding results with the highest probabilities. While these prior-free methods improve localization, they suffer from two primary limitations:

---

[1]Wangxuan Institute of Computer Technology, Peking University. Correspondence to: Yuxin Peng <pengyuxin@pku.edu.cn>.

*Proceedings of the 43$^{rd}$ International Conference on Machine Learning*, Seoul, South Korea. PMLR 306, 2026. Copyright 2026 by the author(s).

(1) *Incompleteness of the search space.* To reduce search complexity, several methods (Wu & Xie, 2024; Shen et al., 2024) partition the original image into discrete grid-based sub-images, which risks fragmenting key semantic regions. Meanwhile, others (Li et al., 2025; Zheng et al., 2025; Zhang et al., 2025b; Lai et al., 2025; Su et al., 2025) rely solely on exploring candidate grounding proposals generated by the model. These proposals may fail to encompass the actual target region, particularly when dealing with long-tail object distributions. Ultimately, these designs impose a ceiling on the achievable performance of search methods.

(2) *Inefficiency due to lack of priors.* Lacking initial semantic guidance from the image content as prior, these methods often require a substantial number of steps to establish a sufficiently informative posterior distribution, resulting in low practical efficiency. For instance, ZoomEye often necessitates over ten interaction rounds with the MLLM to obtain high-quality visual input (Zhong et al., 2025). This incurs prohibitive computational costs that far exceed standard inference, rendering such approaches impractical for many real-world applications.

*Prior-driven Visual search methods* utilize embedding similarity or internal MLLM attention weights as priors salient distribution. Guided by these distributions, they typically employ greedy strategies to select the final visual sub-region. $DC^2$ (Wang et al., 2025b) decomposes high-resolution images into small-scale crops for captioning, subsequently using textual similarity to identify regions most relevant to the query. Similarly, RAP (Wang et al., 2025d) leverages CLIP-based image-text similarity to directly retrieve the best-matching visual input. Another category of works leverages internal MLLM outputs as priors. FOCUS (Zhong et al., 2025) extracts object nouns (e.g., "dog" or "cat") from the query to construct a prior based on embedding similarity within specific MLLM layers, then selects the optimal sub-region by traversing the area of maximum activation. ViCrop (Zhang et al., 2025a) utilizes the attention weights of the last token from a specific layer as a prior, directly selecting the region that maximizes the relative attention weight. Although these methods improve search efficiency, they exhibit two primary deficiencies:

(1) *Loss of logical reasoning context.* These priors fail to capture the abstract logical reasoning information in the query. For instance, extracting object nouns behavior would explicitly discard logical terms. Furthermore, CLIP-based or embedding-based metrics often yield higher similarity scores between "the antonym of dog" and "dog" than with "cat," across both image and text modalities. Additionally, recent research indicates that single-layer attention weights in MLLMs typically fail to capture the complex semantic reasoning capabilities that emerge in deeper layers (Yu & Lee, 2025). As a result, these methods often maintain ob-

jects context but lose the logical reasoning from the query.

(2) *Over-reliance on priors.* By strictly restricting the search range to high-prior regions, these methods become highly susceptible to noise and fluctuations in the prior distribution. If the prior fails to cover important regions or contains sharp noise, the search is prone to converging to local optima.

To address these challenges, we propose **BVS** (**B**ayesian **V**isual **S**earch), a framework that reconceptualizes visual search as a principled global optimization of visual relevance over a continuous spatial-scale manifold. Unlike existing methods that are confined to discrete tiling or static heuristics, BVS establishes a self-correcting search loop that synergizes reasoning-aware priors with iterative posterior updates. Specifically, we navigate the search space by first extracting a reasoning-aware prior via an **Early-stop Attention Rollout**, which preserves the complex logical constraints often lost in shallow feature maps. This prior serves as the initial belief for a Gaussian Process-based active search, where the GP-UCB acquisition function strategically balances the exploration of unobserved regions with the exploitation of high-relevance cues. To navigate the inherent non-stationarity across zoom levels, we introduce a **Scale-Aware Non-stationary Kernel**. This covariance structure explicitly accounts for the varying spatial correlation lengths at different resolutions. By unifying these components, BVS establishes a principled synergy between reasoning-driven prior guidance and observation-based posterior correction. This allows the framework to not only accelerate search through internal model knowledge but also robustly rectify initial biases via direct visual feedback. Furthermore, we provide theoretical foundations that the proposed kernel satisfies *Mercer's theorem*, and establish that BVS achieves a *sub-linear regret bound* under specific assumption, providing a formal guarantee for the convergence and efficiency of the optimization process.

The primary contributions of this work are as follows:

- A Bayesian Optimization-based visual search framework, **BVS**, that simultaneously leverages prior distributions for search acceleration and posterior updates for error correction.

- A **Scale-Aware Non-stationary Kernel** that explicitly captures scale-dependent spatial correlations to accommodate varying visual resolutions and facilitate rapid convergence in multi-scale spaces.

- Beyond empirical results, theoretical guarantees is provided that the proposed kernel satisfies *Mercer's theorem* and the framework achieves *a sub-linear regret bound* under specific assumptions.

## 2. Related Works

**High-Resolution Multimodal Large Language Models.** Recent advancements in Multimodal Large Language Models (MLLMs) have significantly bridged the gap between visual perception and linguistic reasoning (Bai et al., 2025c; Li et al., 2024; Team et al., 2025). Leading open-source models, such as LLaVA-Next (Liu et al., 2024), InternVL (Chen et al., 2024c; Wang et al., 2025c), and Qwen-VL (Bai et al., 2023; 2025a), employ dynamic resolution strategies to handle varying aspect ratios and image details. Despite these architectural optimizations, processing ultra high resolution (UHR) images inevitably introduces a massive number of irrelevant visual tokens, resulting in suboptimal performance on fine-grained tasks requiring the identification of tiny objects in cluttered environments (Wu & Xie, 2024; Lai et al., 2025).

**Learning-based Active Perception.** To overcome resolution bottlenecks, a growing body of work focuses on instilling "active perception" capabilities directly into MLLMs through specialized training. These methods treat visual search as a learnable policy. For instance, SEAL (Wu & Xie, 2024) integrates a distinct scoring head via Supervised Fine-Tuning (SFT) to guide hierarchical exploration. More recently, Reinforcement Learning (RL) has been adopted to train agents that sequentially query visual information, as seen in DeepEyes (Zheng et al., 2025), Thyme (Zhang et al., 2025b), and PixelReasoner (Su et al., 2025). While these learning-based approaches demonstrate strong performance within their training domains, they incur substantial data curation and training costs. Furthermore, their learned policies heavily rely on the precision of intermediate visual grounding. Consequently, they are susceptible to performance degradation in long-tail scenarios where the model fails to ground the target object effectively. In contrast, our approach is entirely training-free, leveraging the bayesian optimization to search the continuous spatial-scale space.

**Inference-time Visual Refinement.** An alternative paradigm focuses on optimizing visual inputs during the inference phase, obviating the need for parameter updates. Early works like $DC^2$ (Wang et al., 2025b) and RAP (Wang et al., 2025d) employ retrieval-based mechanisms, using CLIP embeddings or text similarities to select relevant image crops. More sophisticated methods structure the image into a hierarchical tree or grid, applying search algorithms such as Monte Carlo Tree Search (MCTS) in DyFo (Li et al., 2025) or heuristic traversal in ZoomEye (Shen et al., 2024). Others, such as ViCrop (Zhang et al., 2025a) and FOCUS (Zhong et al., 2025), exploit internal attention maps as saliency indicators. However, a common limitation of these methods is their reliance on *discrete* search spaces, treating the image as a collection of disjoint patches and *greedy* selection strategies. This discretization often frag-

ments semantic information across patch boundaries, while greedy algorithms are prone to local optima. Distinct from these heuristic approaches, BVS formulates visual search as a global optimization problem over a *continuous* spatial-scale manifold. By adopting Bayesian Optimization, we theoretically balance exploration and exploitation, ensuring robust convergence to fine-grained details efficiently.

## 3. Methodology

In this section, we present the problem definition and overall workflow in Section 3.1, followed by the logical reasoning-aware prior acquisition in Section 3.2. We then detail the core Bayesian Optimization (BO) mechanism and our proposed Scale-Aware Non-stationary Kernel in Sections 3.3 and 3.4. Theoretical convergence analysis is provided in Section 3.5, and the final visual consolidation strategy for MLLM is described in Section 3.6.

### 3.1. System Overview

We model the search process as the optimization of an unknown relevance function $f : \mathcal{X} \to [0, 1]$. The search space $\mathcal{X} = \mathcal{S} \times \Omega$ is a continuous manifold spanned by state vectors $\mathbf{z} = (s, \mathbf{x})$, where $s \in \mathcal{S}$ represents the observation scale and $\mathbf{x} \in \Omega \subset \mathbb{R}^2$ denotes the spatial coordinates. As illustrated in Figure 1, the pipeline comprises three stages:

1. **Prior Attention Acquisition**: Takes the query $Q$ and image $\mathcal{I}$ as inputs. By leveraging an attention rollout mechanism with early stopping, it extracts a coarse-grained relevance map from internal attention weights. The output is a non-uniform prior mean function $\mu_0(\mathbf{z})$, initializing the belief over $\mathcal{X}$ before fine-grained exploration.

2. **Active Search via Scale-Aware BO**: Takes $\mu_0$ and $\mathcal{X}$ to drive the search. In each iteration $t$, it updates the Gaussian Process posterior $f \sim \mathcal{GP}(\mu_t, k)$ using the observation history $\mathcal{D}_t$. The module selects an optimal observation point $\mathbf{z}_{t+1}$ via acquisition function maximization and queries the MLLM to obtain a relevance score $y_{t+1} \in [0, 1]$. The output is the converged posterior distribution $(\mu_T, \sigma_T^2)$.

3. **Information Consolidation**: Takes the posterior $\mu_T$ and original image $\mathcal{I}$. It spatially aggregates high-probability regions from $\mu_T$ to filter out background noise. The final output is a consolidated, information-dense visual representation $\mathcal{I}_{final}$ for downstream inference.

### 3.2. Early-stop Attention Rollout

To construct an informative prior mean function $\mu_0(\mathbf{z})$, we leverage the intrinsic cross-modal alignment capabilities of the MLLM. Specifically, we aim to quantify the contribu-

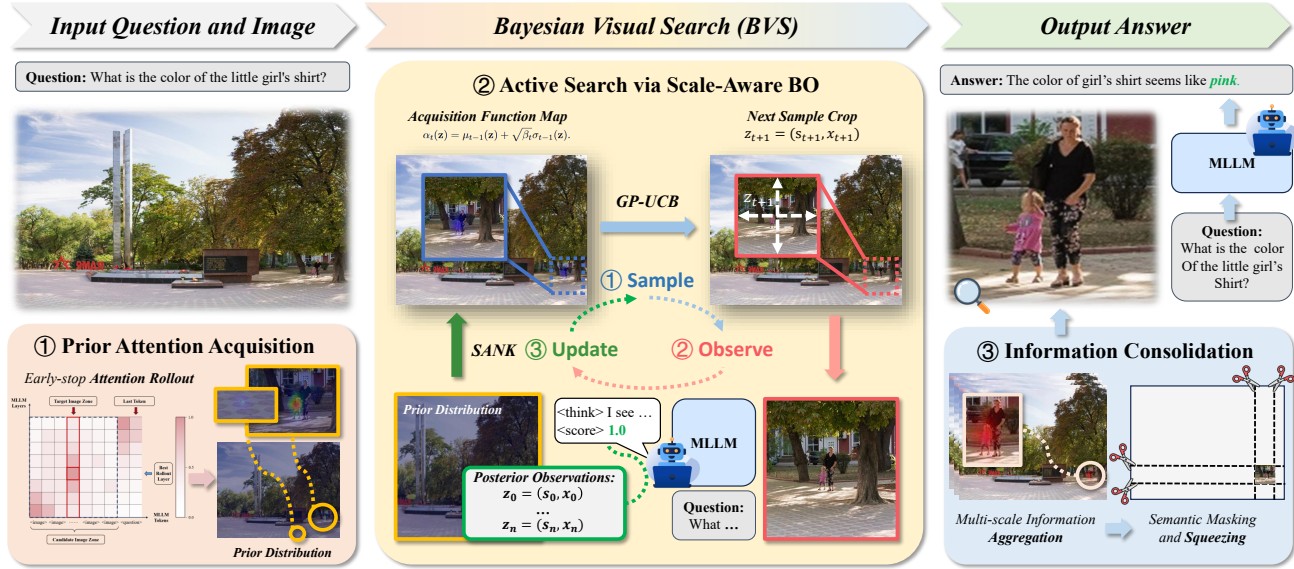

Figure 1. **Overview of the BVS.** Our framework consists of three main stages: (1) **Prior Attention Acquisition**: extracting coarse regions of interest using early-stop attention rollout. (2) **Active Search via Scale-Aware BO**: an iterative Bayesian Optimization loop that progressively locates fine-grained visual evidence. It employs a Scale-Aware Non-stationary Kernel (SANK) to model multi-scale spatial dependencies and a GP-UCB acquisition function to balance exploration and exploitation. (3) **Information Consolidation**: aggregating posterior distribution and performing semantic masking and squeezing to construct a high-utility visual input for the MLLM to generate the final answer.

tion of each visual token to the logical reasoning process embedded in the deep layers of the model.

Let $\mathbf{A}^{(l)} \in \mathbb{R}^{N \times N}$ denote the row-normalized attention matrix at the $l$-th layer, where $N$ is the total number of tokens. We employ an attention rollout strategy to propagate relevance from the final query token back to the visual input. The relevance vector $\mathbf{r}^{(l)}$ at layer $l$ is updated recursively via:

$$\mathbf{r}^{(l-1)} = \frac{1}{2}\left(\mathbf{r}^{(l)}\mathbf{A}^{(l)} + \mathbf{r}^{(l)}\right), \quad (1)$$

where the factor $\frac{1}{2}$ accounts for the skip connections in the architecture, averaging the attention weights and the identity mapping. The recursion is initialized at the last layer $L$ with $\mathbf{r}^{(L)}$ set as a one-hot vector corresponding to the last token.

**The Phenomenon of Attention Divergence.** A direct application of the original Attention Rollout, which propagates relevance through all layers down to the input (i.e., $l \to 0$), proves suboptimal for deep MLLMs in practice Table 4. We observe a phenomenon we term *Attention Divergence*: as relevance propagates into shallower layers, image tokens attention distribution tends to over-smooth and diverge like Figure 4. This contradicts observations in shallower discriminative models (e.g., BERT-based architectures) where full propagation is beneficial (Abnar & Zuidema, 2020).

To mitigate this, we introduce an Early-stop mechanism. We terminate the recursion at the very middle layer $l_{stop} =$

$l_{full}/2$, discarding the noisy signals from shallow layers to avoid attention divergence. The final relevance scores corresponding to the image token indices are then extracted from $\mathbf{r}^{(l_{stop})}$ and reshaped into a 2D heatmap. This heatmap is normalized to $[0, 1]$ to serve as the initial prior $\mu_0(\mathbf{z})$ for the subsequent Bayesian Optimization.

### 3.3. Scale-Aware Non-stationary Covariance

A fundamental challenge in visual search is that spatial correlation varies with observation scale. In a coarse view with large scale, a small spatial displacement has a negligible effect on the window's content. However, in a fine-grained view, the narrow field of view makes the relevance score highly sensitive to minor displacements, as the target can easily exit the crop boundary. To consider this difference, we propose a **Scale-Aware Non-stationary Kernel**.

Let $\ell(s) : \mathcal{S} \to \mathbb{R}^+$ be a positive monotonically increasing function mapping the scale $s$ to a spatial length-scale.

**Definition 3.1** (Scale-Aware Kernel). For any two points $\mathbf{z}_i = (s_i, \mathbf{x}_i)$ and $\mathbf{z}_j = (s_j, \mathbf{x}_j)$ in $\mathcal{X}$, the covariance function $k(\mathbf{z}_i, \mathbf{z}_j)$ is defined as:

$$k(\mathbf{z}_i, \mathbf{z}_j) = \sigma_f^2 \left(\frac{2\ell_i\ell_j}{\ell_i^2 + \ell_j^2}\right) \Psi_\nu \left(\sqrt{\frac{2\|\mathbf{x}_i - \mathbf{x}_j\|^2}{\ell_i^2 + \ell_j^2} + \frac{|s_i - s_j|^2}{\lambda_s^2}}\right), \quad (2)$$

where $\ell_i = \ell(s_i)$, $\Psi_\nu$ is the Matérn radial function, $\sigma_f^2$ is signal variance, and $\lambda_s$ is the $s$ dimension length-scale.

*Table 1.* Comparison results with training-free visual search methods on fine-grained perception benchmarks. The best performance in each task is shown in **bold**, the second-best performance is underlined.

| Model | Method | $V^*$ Bench | | | HR-Bench 4K | | | HR-Bench 8K | | |
|---|---|---|---|---|---|---|---|---|---|---|
| | | *Attr.* | *Spat.* | *Over.* | *FSP* | *FCP* | *Over.* | *FSP* | *FCP* | *Over.* |
| InternVL3.5-4B (Wang et al., 2025c) | w/o reasoning | 67.0 | 55.3 | 62.3 | 56.5 | 42.5 | 49.5 | 51.0 | 40.0 | 45.5 |
| | ZoomEye (2024) *(EMNLP'25)* | 80.9 | 77.6 | 79.6 | 78.0 | 46.0 | 62.0 | 77.5 | 44.5 | 61.0 |
| | RAP (2025d) *(ICML'25)* | 80.0 | 71.1 | 76.4 | 77.5 | 47.8 | 62.6 | 77.0 | 43.8 | 60.4 |
| | ViCrop (2025a) *(ICLR'25)* | 77.4 | 72.4 | 75.4 | 80.3 | 48.3 | 64.3 | 78.0 | 44.8 | 61.4 |
| | BVS (Ours) | **83.5** | **80.3** | **82.2** | **83.5** | **52.3** | **67.9** | **85.0** | 48.8 | **66.9** |
| Qwen3-VL-4B (Bai et al., 2025a) | w/o reasoning | 80.0 | 76.3 | 78.5 | 89.0 | 70.3 | 79.6 | 85.3 | 64.0 | 74.6 |
| | ZoomEye (2024) *(EMNLP'25)* | 91.3 | 81.6 | 87.4 | 91.3 | 72.8 | 82.1 | 89.8 | 64.3 | 77.1 |
| | RAP (2025d) *(ICML'25)* | 87.9 | 80.3 | 84.8 | 91.0 | 68.8 | 79.9 | 86.3 | 56.5 | 71.4 |
| | ViCrop (2025a) *(ICLR'25)* | 85.2 | 77.6 | 82.2 | 89.3 | 71.0 | 80.1 | 85.3 | 62.8 | 74.1 |
| | BVS (Ours) | **96.5** | **82.9** | **91.1** | **91.5** | **73.3** | **82.4** | **93.8** | **65.3** | **79.5** |
| InternVL3.5-8B (Wang et al., 2025c) | w/o reasoning | 65.2 | 72.4 | 68.1 | 70.5 | 56.8 | 63.6 | 60.3 | 53.8 | 57.0 |
| | ZoomEye (2024) *(EMNLP'25)* | 85.2 | 80.3 | 83.2 | 84.3 | 58.3 | 71.3 | 78.5 | 55.5 | 67.0 |
| | RAP (2025d) *(ICML'25)* | 82.6 | 81.6 | 82.2 | 80.0 | 57.5 | 68.8 | 76.8 | 55.3 | 66.1 |
| | ViCrop (2025a) *(ICLR'25)* | 80.0 | 76.3 | 78.5 | 89.5 | 58.5 | 74.0 | 84.0 | 54.3 | 69.2 |
| | BVS (Ours) | **93.0** | **89.5** | **91.6** | **94.8** | **60.5** | **77.6** | **90.8** | **58.5** | **74.6** |
| Qwen3-VL-8B (Bai et al., 2025a) | w/o reasoning | 82.6 | 82.9 | 82.7 | 92.0 | 63.8 | 77.9 | 85.5 | 60.1 | 72.8 |
| | ZoomEye (2024) *(EMNLP'25)* | 92.2 | 86.8 | 90.1 | 92.5 | 65.0 | 78.8 | 88.5 | 62.5 | 75.5 |
| | RAP (2025d) *(ICML'25)* | 88.7 | 88.2 | 88.5 | 91.5 | 60.3 | 75.9 | 86.5 | 58.5 | 72.5 |
| | ViCrop (2025a) *(ICLR'25)* | 86.1 | 82.9 | 84.8 | 93.0 | 64.0 | 78.5 | 86.5 | 61.5 | 74.0 |
| | BVS (Ours) | **95.7** | **89.5** | **93.2** | **96.8** | **72.8** | **84.8** | **95.8** | **63.3** | **79.5** |

This construction allows the "influence radius" of an observation to expand dynamically. When the model performs a small scale observation, the effective spatial length-scale $\ell(s)$ increases to reflect the higher correlation between neighboring pixels in the upscaled feature space.

**Theorem 3.2** (Mercer's Condition). *The scale-aware kernel $k$ defined above is positive semi-definite (PSD) for any set of points in $\mathcal{X}$ and satisfies Mercer's Theorem.*

*Proof Sketch.* The proof is grounded in the generalized non-stationary covariance framework established by Paciorek and Schervish (Paciorek & Schervish, 2006). We represent the search space as a joint 3D manifold $\mathcal{Z} = \mathbb{R}^2 \times \mathbb{R}$ that embeds both spatial and scale dimensions. By specifying a spatially-varying, block-diagonal covariance matrix $\Sigma(\mathbf{z})$ over $\mathcal{Z}$, where the spatial components are conditioned on the scale coordinate, we ensure the resulting kernel satisfies the necessary conditions for positive semi-definiteness. Detailed derivations are provided in Section B.1. □

*Remark* 3.3 (Implementation). In practice, we use the arithmetic mean $L_{ij} = (\ell(s_i) + \ell(s_j))/2$ as a first-order approximation of the Gibbs length-scale for computational efficiency. Empirical results show that this maintains the stability of the Cholesky decomposition throughout the optimization process.

### 3.4. Bayesian Optimization for Visual Search

This section details how we integrate the attention prior from Section 3.2 and the scale-aware kernel from Section 3.3 into the sequential decision process.

**Prior-Informed Mean Function.** Unlike standard zero-mean GPs, we leverage the global context from the MLLM. Utilizing the attention heatmap $\mathbf{A}$ obtained via Early-stop Attention Rollout (Section 3.2), we define the continuous *prior mean function* $\mu_0(\mathbf{z})$ for any $\mathbf{z} = (s, \mathbf{x}) \in \mathcal{X}$ as:

$$\mu_0(s, \mathbf{x}) = \theta \cdot \text{Interp}(\mathbf{A}, \mathbf{x}), \qquad (3)$$

where $\text{Interp}(\cdot)$ maps the discrete attention grid to the continuous spatial domain $\Omega$. $\mu_0$ is initially assumed invariant across the scale $s$, with $\theta$ modulating the prior's influence.

**MLLM Evaluation.** In each iteration $t$, given a candidate $\mathbf{z}_t = (s_t, \mathbf{x}_t)$, we extract the corresponding image crop and query the MLLM. The MLLM acts as a scoring oracle, returning a scalar relevance score $y_t \in [0, 1]$ based on the semantic alignment between the crop and the query. We model this as a noisy observation $y_t = f(\mathbf{z}_t) + \epsilon$, where $\epsilon \sim \mathcal{N}(0, \sigma_n^2)$.

**Acquisition via GP-UCB.** To balance the exploration of unobserved scales and the exploitation of high-relevance regions, we employ the Gaussian Process Upper Confidence Bound (GP-UCB) acquisition function (Srinivas et al., 2009):

$$\alpha_t(\mathbf{z}) = \mu_{t-1}(\mathbf{z}) + \sqrt{\beta_t}\sigma_{t-1}(\mathbf{z}). \qquad (4)$$

*Table 2.* Comparison results with training-based visual search on fine-grained perception benchmarks. The best performance in each task is shown in **bold**, the second-best performance is underlined.

| Method | $V^*$ **Bench** | | | **HR-Bench 4K** | | | **HR-Bench 8K** | | |
|---|---|---|---|---|---|---|---|---|---|
| | *Attribute* | *Spatial* | *Overall* | *FSP* | *FCP* | *Overall* | *FSP* | *FCP* | *Overall* |
| Pixel Reasoner-7B (Su et al., 2025) | - | - | 84.3 | - | - | - | - | - | - |
| Simple O3-7B (Wang et al., 2025e) | - | - | 90.4 | - | - | 76.2 | - | - | - |
| DeepEyes-7B (Zheng et al., 2025) | 91.3 | 88.2 | 90.1 | 91.3 | 59.0 | 75.1 | 86.8 | 58.5 | 72.6 |
| Thyme-7B (Zhang et al., 2025b) | 83.5 | 80.3 | 82.2 | 91.0 | 63.0 | 77.0 | 86.5 | 57.5 | 72.0 |
| TreeVGR-7B (Wang et al., 2025a) | 94.0 | 87.0 | 91.1 | 90.3 | 64.0 | 77.1 | 86.5 | 59.8 | 73.1 |
| Mini-o3-7B (Lai et al., 2025) | - | - | 88.2 | - | - | 77.5 | - | - | 73.3 |
| Qwen3-VL-8B-*w tool* (Bai et al., 2025a) | - | - | 90.1 | - | - | 82.3 | - | - | 78.0 |
| BVS-Qwen3VL-8B (Ours) | **95.7** | **89.5** | **93.2** | **96.8** | **72.8** | **84.8** | **95.8** | **63.3** | **79.5** |

The posterior mean $\mu_t$ and variance $\sigma_t^2$ are updated using the standard GP regression identities:

$$\begin{aligned}
\mu_t(\mathbf{z}) &= \mu_0(\mathbf{z}) + \mathbf{k}_t(\mathbf{z})^T(\mathbf{K}_t + \sigma_n^2\mathbf{I})^{-1}(\mathbf{y}_t - \boldsymbol{\mu}_{0,t}), \\
\sigma_t^2(\mathbf{z}) &= k(\mathbf{z},\mathbf{z}) - \mathbf{k}_t(\mathbf{z})^T(\mathbf{K}_t + \sigma_n^2\mathbf{I})^{-1}\mathbf{k}_t(\mathbf{z}),
\end{aligned} \quad (5)$$

where $\mathbf{k}_t(\mathbf{z}) = [k(\mathbf{z}_1,\mathbf{z}),\ldots,k(\mathbf{z}_t,\mathbf{z})]^T$ is the covariance vector, and $\mathbf{K}_t$ is the $t \times t$ scale-aware covariance matrix. The exploration-exploitation trade-off parameter $\beta_t$ is chosen to satisfy regret bound conditions, typically increasing logarithmically with $t$ to facilitate broader exploration as the search budget is consumed.

### 3.5. Convergence and Regret Analysis

Let $\mathbf{z}^* = \arg\max_{\mathbf{z}\in\mathcal{X}} f(\mathbf{z})$ be the optimal observation window. We analyze the cumulative regret $R_T = \sum_{t=1}^{T}[f(\mathbf{z}^*) - f(\mathbf{z}_t)]$ and aim to show that $\lim_{T\to\infty} R_T/T = 0$.

**Assumption 3.4** (RKHS Norm with Prior). We assume the residual function $\tilde{f}(\mathbf{z}) = f(\mathbf{z}) - \mu_0(\mathbf{z})$ resides in the Reproducing Kernel Hilbert Space (RKHS) $\mathcal{H}_k$ associated with the scale-aware kernel $k$, with the bounded norm $\|\tilde{f}\|_k \leq B$.

**Theorem 3.5** (Scale-Aware Regret Bound). *Under Assumption 3.4, let the exploration parameter be $\beta_t = 2B^2 + 300\gamma_t\log^3(t/\delta)$. With probability at least $1 - \delta$, the cumulative regret of* BVS *satisfies:*

$$R_T \leq \sqrt{C \cdot T\beta_T\gamma_T}, \quad (6)$$

*where $C = 8/\log(1 + \sigma_n^{-2})$ and $\gamma_T$ is the Maximum Information Gain (MIG) of the scale-aware kernel $k$ after $T$ observations.*

*Proof.* **Residual GP Consistency**: By formulating the GP over the residual $\tilde{f}$, the observation $R_t - \mu_0(\mathbf{z}_t)$ provides a noisy estimate of $\tilde{f}(\mathbf{z}_t)$. Since $k$ is a valid PSD kernel (Theorem 3.1), the confidence intervals constructed via $\mu_t \pm \sqrt{\beta_t}\sigma_t$ remain well-calibrated for the target function $f$ with high probability (Srinivas et al., 2009).

**MIG under Scale-Dependent Length-scales**: In our search space, the spatial length-scale $\ell(s)$ is bounded as $\ell_{\min} \leq \ell(s) \leq \ell_{\max}$. Since the domain $\mathcal{X}$ is compact and $k$ is locally equivalent to a stationary Matérn kernel, the eigenvalues of the covariance matrix $\mathbf{K}_T$ decay at a polynomial rate. For $d = 3$, this yields a sub-linear MIG growth $\gamma_T = \mathcal{O}(T^{\frac{d(d+1)}{2\nu+d(d+1)}}\log T)$, ensuring $R_T/T \to 0$ as $T \to \infty$.

**Prior Acceleration**: The constant factor $B = \|f - \mu_0\|_k$ reflects the initial knowledge gap. An informative MLLM prior ensures $\|f - \mu_0\|_k \ll \|f\|_k$, effectively shrinking the search volume in the early iterations of the BO process. $\square$

### 3.6. Information Consolidation

The final search yields a posterior $\mu_T(s, \mathbf{x})$ characterizing relevance across the spatial-scale manifold. To generate visual content suitable for direct MLLM input, we consolidate disjoint yet critical regions into a compact representation.

**Multi-scale Aggregation.** We first aggregate $\mu_T$ across the scale dimension $s$ to generate a unified 2D spatial importance map $M(\mathbf{x}) = \max_{s\in\mathcal{S}} \mu_T(s, \mathbf{x})$. This ensures that relevance detected at any zoom level is preserved.

**Semantic Masking and Squeezing.** A binary mask $\mathcal{B}(\mathbf{x})$ is derived by applying an adaptive threshold $\tau$ to $M(\mathbf{x})$:

$$\mathcal{B}(\mathbf{x}) = \mathbb{K}[M(\mathbf{x}) > \tau]. \quad (7)$$

Based on $\mathcal{B}(\mathbf{x})$, the *Squeezing* operation removes redundant background pixels. Unlike standard rectangular cropping, our algorithm identifies the minimal set of rows $\mathcal{R}$ and columns $\mathcal{C}$ that intersect with the support of $\mathcal{B}(\mathbf{x})$. The final squeezed image $I_{sq}$ is constructed as the sub-matrix:

$$I_{sq} = I[\mathcal{R}, \mathcal{C}]. \quad (8)$$

This process effectively "collapses" the uninformative space between distant relevant objects, resulting in a compact, information-dense composite representation.

# 4. Experiments

## 4.1. Setups

**Benchmarks and Metrics.** Following prior work (Wu & Xie, 2024; Wang et al., 2025d;b; Zhong et al., 2025; Zhang et al., 2025a), we evaluate BVS on commonly used fine-grained perception benchmarks: (1) $V^*$ **Bench** (Wu & Xie, 2024), composed of the attribute recognition and spatial reasoning two categories, with an average image resolution of $[2246.27, 1582.98]$; (2) **HRBench4K** and **HRBench8K** (Wang et al., 2025b), covering single-object and cross-object perception, with an average image resolution of $[4023.93, 3502.94]$ and $[7431.12, 5357.56]$; To assess performance on standard-resolution tasks, we also include widely taken standard benchmark **MMStar** (Chen et al., 2024a). We report accuracy as the primary metric across all benchmarks.

**Baselines.** To validate the generalizability of our method and compare it with state-of-the-art results, we employ the latest MLLM series, specifically the 4B and 8B variants of InternVL3.5 and Qwen3-VL as baselines. We compare BVS against open-source visual search methods, including ZoomEye (Shen et al., 2024), RAP (Wang et al., 2025d), and ViCrop (Zhang et al., 2025a).

**Implementation Details.** We utilize Qwen2.5-VL-3B and Qwen2.5-VL-7B as proxy models to acquire attention priors for the 4B and 8B scale base models, respectively. This proxy-based strategy is adopted to construct an offline attention cache mechanism, significantly accelerating the experimental iteration. According to official reports (Bai et al., 2025b; Wang et al., 2025c; Bai et al., 2025a), these proxy models possess strictly weaker perception and reasoning capabilities compared to the newer InternVL3.5 and Qwen3-VL models of comparable size, ensuring that no unfair prior knowledge should be introduced. All benchmark evaluations are conducted using the VLMevalkit framework (Duan et al., 2024) to ensure a unified and fair comparison. By default, the confidence threshold is set to $\tau = 0.2$, and the maximum number of search steps is limited to 5 unless otherwise specified.

## 4.2. Main Results on Fine-grained Perception

In Table 1 and Table 2, BVS demonstrates robust performance, surpassing other competitive visual search methods across all evaluated fine-grained perception benchmarks. On the $V^*$ benchmark, BVS achieves an overall score of 93.2, outperforming the second-best method, ZoomEye, by $+3.1$ points. This trend is mirrored on HRBench4K and HRBench8K; specifically, for HRBench4K, BVS attains an overall score of 84.8. To the best of our knowledge, this represents the highest result currently reported on this benchmark for an 8B-scale model. These results strongly support

the effectiveness of BVS in fine-grained perception tasks.

## 4.3. Ablation Studies and Efficiency Analysis

In this subsection, we validate the design choices of BVS on $V^*$ Bench, using Qwen3-VL-8B as the base model.

**Analysis of Attention Priors.** We compare our proposed attention rollout strategy against three alternative schemes: (1) *Full-layer attention rollout*: Accumulating attention from the first to the last layer. (2) *Object embedding similarity*: Calculating priors via text-image token embedding cosine similarity. (3) *Intermediate attention weights*: Using only the last token's attention from a fixed intermediate layer.

As shown in Table 4, our method achieves the highest coverage rate (92.9%) and accuracy (93.2%). Notably, full-layer rollout performs poorly (12.0% coverage) because attention distributions in lower layers often drift toward non-image context tokens, diluting the local semantic information. While embedding similarity provides a semantic baseline, we observe that it introduces excessive irrelevant noise, which may stem from the fact that embeddings in MLLMs are not natively optimized for direct similarity computation. In contrast, intermediate attention weights are observed losing attention for partial objects. This may indicate that attention signals for specific objects are distributed across different layers; consequently, relying solely on single-layer weights is insufficient to fully encapsulate the MLLM's comprehensive focus. Our strategy strikes the best balance by capturing high-resolution image cues.

**Kernel Function Analysis.** To verify that our kernel function effectively accelerates search by dynamically adjusting the length scale, we compare the proposed Scale-Aware Non-stationary Kernel against a standard Matérn kernel. As shown in Table 5, BVS reaches an accuracy of 93.2% in only 5 steps, which is already superior to the Matérn kernel's performance at 15 steps (92.7%). While both kernels eventually converge to a similar accuracy at 50 steps, our scale-aware design significantly reduces the search budget required to reach the performance plateau, proving its efficiency in practical deployments.

**Efficiency Analysis.** Following (Zhong et al., 2025), we plot the inference accuracy against the number of forward passes (FPs) in Figure 2. BVS (BVS) achieves a significantly better Pareto frontier compared to existing baselines. Specifically, on HRBench-8K, BVS achieves $> 78\%$ accuracy with fewer than 5 FPs, whereas methods like ZoomEye and RAP require 20 to 35 FPs to achieve lower or comparable results. This confirms that our method is not only more accurate but also drastically more computationally efficient.

*Table 3.* Results on MMStar dataset. We compare our `BVS` with the base Qwen3-VL-8B model across various categories.

| Model | Overall Acc. | Δ | Coarse Perc. Acc. | Δ | Fine-grained Acc. | Δ | Inst. Reason. Acc. | Δ | Logi. Reason. Acc. | Δ | Math Acc. | Δ | Sci & Tech Acc. | Δ |
|---|---|---|---|---|---|---|---|---|---|---|---|---|---|---|
| Qwen3VL-8B | 66.20 | - | 74.00 | - | 56.00 | - | 75.20 | - | 68.00 | - | 70.40 | - | 53.60 | - |
| w/ `BVS` | 66.07 | -0.13 | 76.00 | **+2.00** | 60.00 | **+4.00** | 74.00 | -1.20 | 66.00 | -2.00 | 70.40 | 0.00 | 50.00 | -3.60 |

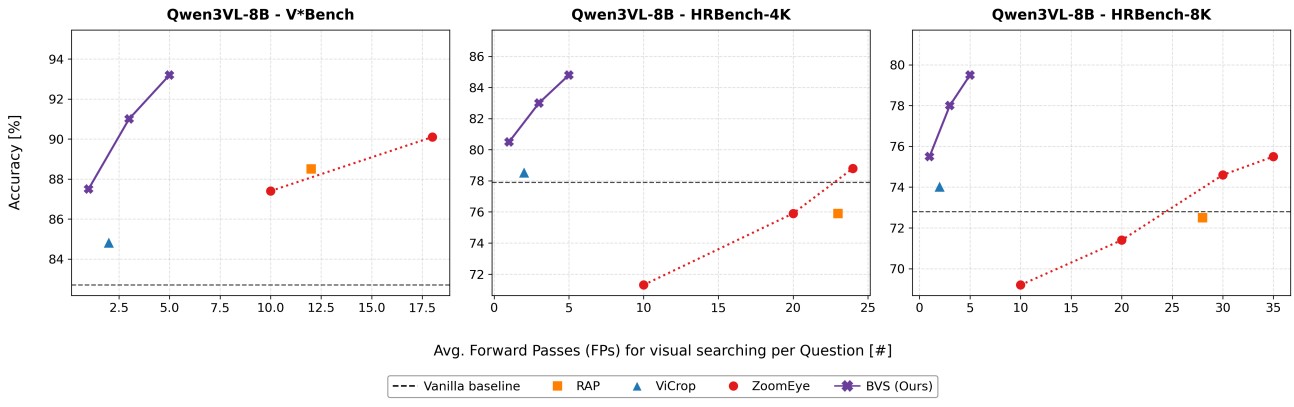

*Figure 2.* Efficiency Analysis of `BVS` comparing with other training-free visual search methods.

*Table 4.* Ablation study on different attention priors using Qwen3-VL-8B. We report the ground-truth region coverage and final accuracy on $V^*$ dataset.

| Attention Prior Strategy | Coverage Rate (%) ↑ | Accuracy (%) ↑ |
|---|---|---|
| (1) Full-layer attention rollout | 12.0 | 82.2 |
| (2) Object embedding similarity | 66.4 | 84.8 |
| (3) Intermediate attention weights | 76.5 | 88.5 |
| **BVS (Ours)** | **92.9** | **93.2** |

*Table 5.* Efficiency and accuracy analysis of kernel functions on $V^*$. We report Accuracy (%) at different search steps $N$.

| Kernel | $V^*$ Accuracy (%) at different steps ↑ | | |
|---|---|---|---|
| | @ 5 Steps | @ 15 Steps | @ 50 Steps |
| Standard Matérn | 90.1 | 92.7 | 93.2 |
| **Scale-Aware (Ours)** | **93.2** | **93.7** | **93.7** |

### 4.4. Additional Results

**Results on General Perception Benchmarks.** While `BVS` is primarily designed for fine-grained perception, we evaluate its generalization capability on standard-resolution tasks using the MMStar benchmark (Table 3). The results show that `BVS` significantly improves performance in fine-grained perception (+4.0%) and coarse perception (+2.0%). Although a marginal decline is observed in logical reasoning, we attribute this to the fact that such tasks are predominantly reasoning-intensive rather than perception-driven; hence, enhancing visual granularity may not bypass the model's inherent cognitive bottlenecks. Overall, the competitive performance across all dimensions demonstrates that `BVS`

acts as a specialized "magnifying glass," substantially bolstering perception without compromising the foundational capabilities of the base MLLM.

### 4.5. Visualization

To provide an intuitive understanding of `BVS`, we visualize two representative examples from the $V^*$ benchmark in Figure 3. The posterior mean maps generated by `BVS` accurately reflect the spatial distribution of relevance across the image relative to the query. In contrast to Qwen3-VL, which relies on discrete coordinate-based crop tools, our approach is more effective at preserving the integrity of critical visual evidence.

As illustrated in the first case (top row), the question requires determining the spatial relationship between a soccer ball and a water dispenser. Qwen3VL $w/$ tool only captures a high-resolution crop of the soccer ball but fails to include the water dispenser, leading to a "contextual void" and an incorrect answer. In contrast, `BVS`, guided by the posterior mean maps, sequentially identifies both key regions (labeled circle 1 and circle 2), providing sufficient visual evidence for the model to reason correctly.

## 5. Conclusion

In this paper, we presented **BVS**, a Bayesian Optimization-based framework designed to enhance the fine-grained perception capabilities of MLLMs in ultra-high-resolution scenarios. `BVS` effectively bridges the gap between prior-free and prior-driven strategies by integrating the **Early-stop**

**Question:** Is the soccer ball on the left or right side of the water dispenser? (A) left  (B) right
**Label:** B | **Answer (Qwen3VL):** A ✗ | **Answer (Qwen3VL *w/ tool*):** A ✗ | **Answer (Qwen3VL *w/ BVS*):** B ✓

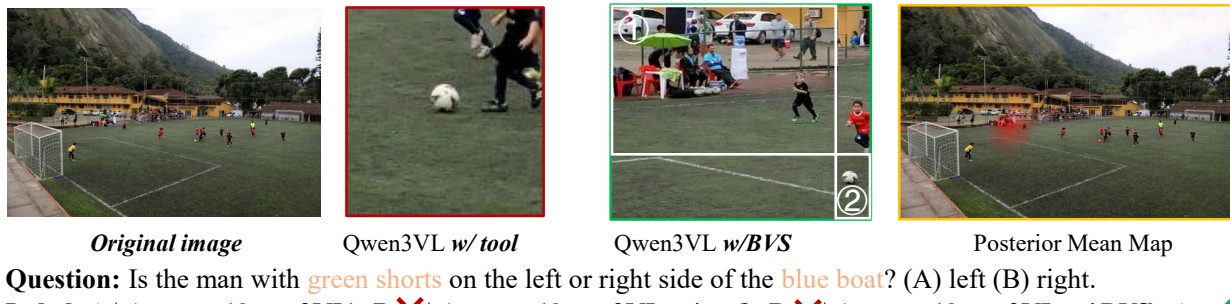

*Original image*      Qwen3VL *w/ tool*      Qwen3VL *w/BVS*      Posterior Mean Map

**Question:** Is the man with green shorts on the left or right side of the blue boat? (A) left (B) right.
**Label:** A | **Answer (Qwen3VL):** B ✗ | **Answer (Qwen3VL *w/ tool*):** B ✗ | **Answer (Qwen3VL *w/ BVS*):** A ✓

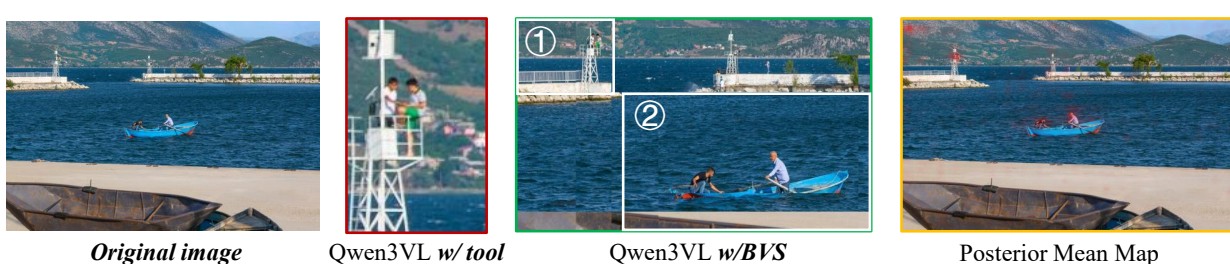

*Original image*      Qwen3VL *w/ tool*      Qwen3VL *w/BVS*      Posterior Mean Map

*Figure 3.* Cases study from V* Bench of `BVS` with Qwen3VL-8B. Qwen3-VL *w/tool* output is also reported to compare.

**Attention Rollout** prior, which captures logical reasoning context with a rigorous posterior correction mechanism. A key contribution is the **Scale-Aware Non-stationary Kernel**, which adapts the sampling strategy to dynamic observation scales, thereby preventing inefficient exploration caused by scale mismatch. We provided theoretical analysis proving the validity of our kernel and the convergence of the optimization process. Empirical results across various benchmarks confirm that `BVS` not only locates tiny objects more accurately but also operates with significantly fewer model interactions than existing methods. We believe this work offers a solid theoretical foundation and a practical solution for efficient, high-resolution visual perception in future multimodal systems.

## Impact Statement

This paper presents work whose goal is to advance the field of Machine Learning. There are many potential societal consequences of our work, none which we feel must be specifically highlighted here.

## Acknowledgements

This work was supported by the grants from the National Natural Science Foundation of China (62525201, 62132001, 62432001) and Beijing Natural Science Foundation (L247006).

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

# Appendix

## A. Visualization of the Phenomenon of Attention Divergence

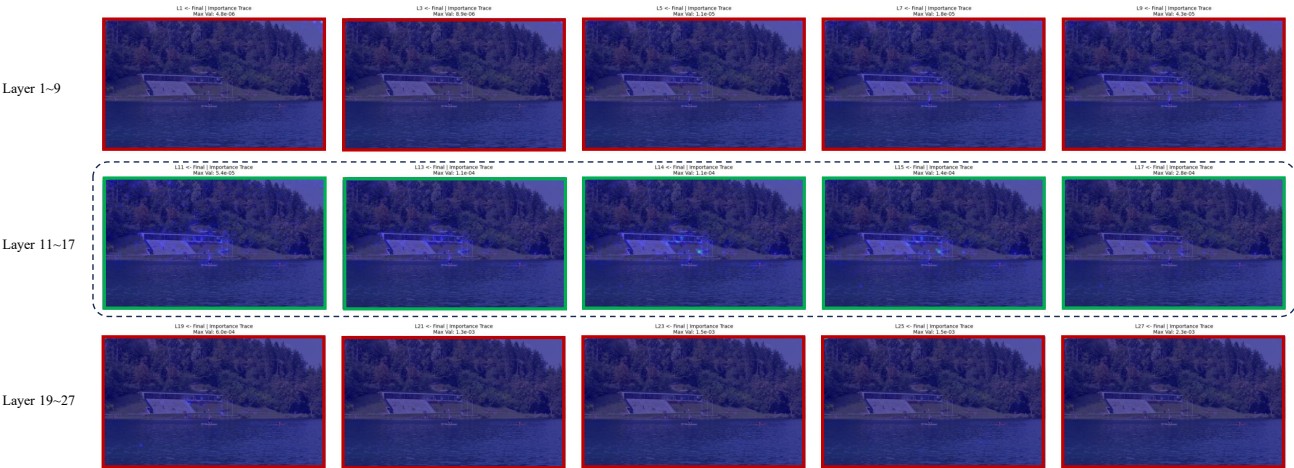

*Figure 4.* **The Phenomenon of Attention Divergence.** An attention rollout case study from V* Bench illustrating the evolution of attention maps across a full 28-layer Qwen2.5VL. Both insufficient depths ($\leq 9$) and excessive depths ($\geq 19$) fail to capture critical visual regions accurately. Relevant visual signals only become prominent within the intermediate layers, highlighting the necessity of our early-stopping mechanism.

In Section 3.2 of the main text, we introduced the concept of *Attention Divergence* to explain why propagating attention weights through the entire network depth is suboptimal for fine-grained localization. To empirically validate this claim and justify our proposed **Early-stop Attention Rollout** mechanism, we provide a detailed visualization of the attention maps across different layers in Figure 4.

## B. Theoretical Analysis

### B.1. Proof of Theorem 3.2 (Positive Definiteness)

**Theorem 3.2.** *The scale-aware kernel $k$ defined in Eq. (2) is positive semi-definite (PSD) for any set of points in $\mathcal{X}$ and satisfies Mercer's Theorem.*

*Proof.* The proof relies on the general framework for non-stationary covariance functions established by Paciorek and Schervish (Paciorek & Schervish, 2006). Their theorem states that for any valid stationary correlation function $k_S(\cdot)$ on $\mathbb{R}^d$ and any mapping $\mathbf{z} \to \Sigma(\mathbf{z})$ where $\Sigma(\mathbf{z})$ is a positive definite $d \times d$ Gaussian covariance matrix at each point, the following function is positive semi-definite:

$$k_{NS}(\mathbf{z}_i, \mathbf{z}_j) = |\Sigma_i|^{\frac{1}{4}} |\Sigma_j|^{\frac{1}{4}} \left| \frac{\Sigma_i + \Sigma_j}{2} \right|^{-\frac{1}{2}} k_S \left( \sqrt{Q_{ij}} \right), \tag{9}$$

where $Q_{ij} = (\mathbf{z}_i - \mathbf{z}_j)^\top \left( \frac{\Sigma_i + \Sigma_j}{2} \right)^{-1} (\mathbf{z}_i - \mathbf{z}_j)$.

In our formulation, we define the search space over the augmented vector $\mathbf{z} = [\mathbf{x}^\top, s]^\top \in \mathbb{R}^3$. We construct a block-diagonal covariance matrix $\Sigma(\mathbf{z})$ that treats the spatial and scale dimensions independently but allows the spatial variance to depend on the scale $s$:

$$\Sigma(\mathbf{z}) = \begin{bmatrix} \ell(s)^2 \mathbf{I}_2 & \mathbf{0} \\ \mathbf{0} & \lambda_s^2 \end{bmatrix}, \tag{10}$$

where $\mathbf{I}_2$ is the $2 \times 2$ identity matrix for the spatial coordinates. Since $\ell(s) > 0$ and $\lambda_s > 0$, $\Sigma(\mathbf{z})$ is positive definite for all $\mathbf{z} \in \mathcal{X}$.

Substituting this structure into the general form:

**1. Pre-factor derivation:** The determinant is $|\Sigma_i| = \ell(s_i)^4 \lambda_s^2$. The determinant of the average matrix is $|\frac{\Sigma_i + \Sigma_j}{2}| = (\frac{\ell_i^2 + \ell_j^2}{2})^2 \lambda_s^2$. The pre-factor simplifies to:

$$\frac{(\ell_i^4 \lambda_s^2)^{1/4}(\ell_j^4 \lambda_s^2)^{1/4}}{\sqrt{(\frac{\ell_i^2 + \ell_j^2}{2})^2 \lambda_s^2}} = \frac{\ell_i \ell_j \lambda_s}{\frac{\ell_i^2 + \ell_j^2}{2} \lambda_s} = \frac{2\ell_i \ell_j}{\ell_i^2 + \ell_j^2}. \tag{11}$$

**2. Distance metric derivation:** The quadratic form $Q_{ij}$ decomposes into spatial and scale components:

$$Q_{ij} = \frac{(\mathbf{x}_i - \mathbf{x}_j)^\top (\mathbf{x}_i - \mathbf{x}_j)}{\frac{\ell_i^2 + \ell_j^2}{2}} + \frac{(s_i - s_j)^2}{\lambda_s^2}. \tag{12}$$

This exactly matches the argument inside the Matérn function in Eq. (2) of the main paper.

Since the Matérn function $\Psi_\nu$ is a valid stationary correlation function on $\mathbb{R}^3$, and our construction satisfies the conditions of Paciorek and Schervish's theorem, the resulting scale-aware Kernel $k$ is positive semi-definite on $\mathcal{X}$. $\qquad\square$

## C. Prompt Details

### C.1. Prompt for MLLM Evaluation as Oracle

As described in Section 3.4, we employ the MLLM as a scoring oracle to evaluate the semantic relevance between a candidate image crop and the input query. To ensure the transparency and reproducibility of our search process, we provide the exact system prompt used for this evaluation below. This prompt is designed to steer the MLLM toward a "Visual Entity Matching" behavior, prioritizing the existence of query-related entities over complex spatial reasoning during the sampling phase. Notably, we consistently apply this standardized prompt across all evaluated benchmarks, including MMStar, to ensure the generalizability of our results.

---

**MLLM Visual Relevance Scoring Prompt**

```
You are a Visual Entity Matching Engine.  Your task is to calculate the intersection
between the nouns in a text query and the objects in an image.
Mandatory Thinking Process In <THOUGHT> (max 3 sentences): 1.  Identify ALL Nouns:
List every physical object mentioned (e.g., "backpack", "clock").  2.  Scan Image:
Search for each noun independently.  3.  Apply the "Positive Presence" Rule:  If any
listed noun is found, the score MUST be above 0.0.
Strict Rules & Scoring Examples Rule 1:  Equal Weighting of Nouns Do not distinguish
between the "subject" and the "reference object." Every noun mentioned is a target.
Example:  Query:  "Is the backpack near the clock?" - If you see ONLY the backpack ->
Score 0.6.  - If you see ONLY the clock -> Score 0.6.  - Reasoning:  One of the two
targets is 100% present.
Rule 2:  Anti-Zero Penalty (Crucial) Assigning 0.0 when at least one entity is
visible is considered a system error.  Even if the question is unanswerable, the
presence of an entity creates visual relevance.
Rule 3:  Scoring Scale - 1.0:  All nouns from the text are clearly visible.  - 0.5
- 0.8:  At least one primary noun is found.  (e.g., 1 out of 2 nouns = 0.6; 1 out of
3 nouns = 0.4).  - 0.1 - 0.4:  No full objects are found, but a related context or a
partial fragment of a target is visible.  - 0.0:  Absolutely none of the mentioned
nouns or their immediate context are in the image.
Rule 4:  Ignore "Action" or "Position" Disregard words like "left," "right," "on,"
"inside," or "chasing." Only score based on the existence of the nouns.
Output Format <THOUGHT> (Brief analysis following the steps above) <SCORE> (Single
float between 0.0 and 1.0)
Input Question:  [Query Text] Analyze and provide the output:
```

---

