# OpenReview forum: "BVS: Bayesian Visual Search with Multimodal Large Language Model for Fine-grained Perception"
_ICML.cc/2026/Conference — ICML 2026 regular_

### Official Review · Reviewer_XB8m · 2026-03-09

**Soundness:** 3
**Presentation:** 3
**Significance:** 3
**Originality:** 3
**Overall Recommendation:** 4
**Confidence:** 3

**Summary:**

The paper targets fine-grained perception in ultra-high-resolution images. Its main argument is that prior-free search is often inefficient and incomplete, while prior-driven search is faster but brittle because it over-trusts static saliency priors. BVS tries to bridge those two regimes by optimizing a relevance function over a continuous spatial-scale manifold: it builds an initial prior via early-stop attention rollout, uses GP-UCB with a scale-aware non-stationary kernel to choose crops, and then consolidates the posterior into a compact image for the final MLLM answer.

**Compliance With Llm Reviewing Policy:**

Affirmed.

**Key Questions For Authors:**

1. Does the efficiency analysis include the cost of proxy prior generation? You use Qwen2.5-VL proxy models plus an offline attention cache, while Figure 2 reports forward-pass efficiency. What is the true end-to-end latency per query if prior generation is counted?
2. How is the acquisition function optimized in practice? Since the method is posed over a continuous spatial-scale manifold, what is the actual optimizer for \alpha_t(z)? Random restarts, discretized candidates, gradient-based search, or something else? What is its computational complexity?
3. Do the theoretical guarantees apply to the implementation you evaluate? More specifically, do they still hold when using the arithmetic-mean kernel approximation and the prompt-engineered entity-matching oracle rather than an idealized noisy evaluation of f?
4. Can you validate the “reasoning-aware prior” claim more directly? Right now, the search prompt mostly rewards noun coverage. Can you show controlled experiments on negation, compositional logic, or relation-heavy queries where noun sets are the same but logic differs?

**Limitations:**

Please refer to weakness and questions.

**Strengths And Weaknesses:**

### Strengths
1.	Well-motivated problem setup. The paper clearly explains the gap between prior-free methods and prior-driven methods, and that framing makes the proposed hybrid approach easy to understand. The problem itself—searching tiny, query-relevant evidence in cluttered UHR images—is important and timely.
2. The three-stage pipeline is coherent and pretty principle from human understanding: attention-derived prior, BO-based active search, then posterior-based consolidation. Framing search over a continuous spatial-scale space is more principled than purely heuristic discrete crop selection.
3. The gains on V* and HRBench look pretty good. The SOTA results evidented the paper’s contribution.
4. Table 4 supports early-stop rollout over full rollout, object-embedding similarity, and single-layer attention. Table 5 suggests the scale-aware kernel improves convergence speed, not just final accuracy.
5. The writing and figures are pretty good.
6. The paper does not just present a heuristic; it tries to justify the kernel and BO procedure formally. Even if I have concerns about the final match between theory and implementation, this is still a plus.

### Weaknesses
1. The “reasoning-aware” claim is under-supported.
The paper repeatedly claims that early-stop attention preserves logical reasoning context, but the actual sampling prompt ignores words like left/right/on/inside, and the MMStar results do not show a general reasoning benefit. That makes the reasoning claim feel stronger than the evidence.
2. The implementation uses Qwen2.5-VL proxy models and an offline attention cache, while Figure 2 reports forward-pass efficiency. It is not clear whether the cost of generating those priors is included in the FP counts.
3. Some implementation details remain underspecified. Section 2.4 gives the GP-UCB equations, but I could not find a clear practical description of how the acquisition function is optimized over the continuous space X, or how sensitive results are to the scale parameterization. There is also a mild inconsistency in the kernel intuition: \ell(s) is defined as increasing with scale, but the prose later says it increases for “small scale” observations.

---

> ### Author Rebuttal · Authors · 2026-03-31
>
> **Response to Reviewer XB8m.**
>
> We thank you for the insightful reviews. We are encouraged that you find the problem well motivated, the three-stage pipeline coherent, and the results on V* and HRBench convincing. We also appreciate your recognition of Table 4–5 in supporting the early-stop prior and scale-aware kernel, as well as our effort to provide a formal justification of the BO framework. We address your questions and concerns below.
>
> **W1 & Q4: Clarification and validation of the “reasoning-aware prior”.**
>
> **A1:** We specifically refer "reasoning-aware" to the **Prior Attention Acquisition stage**, where search is guided by the MLLM’s cross-modal understanding of the query’s compositional structure.
>
> To clarify:
> 1. **Search prior vs. sampling oracle:** The noun-focused prompt is designed to maximize **recall during search**. Since a high-resolution crop may contain only one relevant entity, the relational constraints is prone to assigning low scores and discard useful regions. More analysis see rebuttal to *GLbc* in A3.
>
> 2. **Winoground validation:** We evaluate on **Winoground**, where queries share identical nouns but differ in relations (e.g., “book on cup” vs. “cup on book”). A noun-only prior (CLIP) achieves **50.4%** (near random), while **BVS reaches 74.2%** accuracy, demonstrating that the prior adapts to relational context beyond a bag-of-nouns signal.
>
> 3. **MMStar results:** BVS primarily enhances **perception quality** by providing better visual evidence. More analysis see rebuttal to *bRSs* in A3.
>
>
> **W2 & Q1: Inclusion of proxy prior generation cost in efficiency metrics.**
>
> **A2:** We appreciate this clarification. To be transparent, Figure 2 reports only the cost of active search iterations ($T$). Since we used an offline cache, the one-time prior generation (≈ **1 FP**) was not included in that plot. To provide a complete view, we include this initialization cost in [figure](https://anonymous.4open.science/r/anonymous-90FC/vstarbench_three_panel_efficiency.png), even with prior generation, BVS remains significantly more efficient:
> 1. **Total FP Count:** A 5-step BVS requires **6 FPs (1 prior + 5 steps)**, vs. ZoomEye (18) and RAP (12).
> 2. **Total Compute (FLOPs):** Prior generation is lightweight (~102 TFLOPs), and each step costs ~12.7 TFLOPs, totaling **165 TFLOPs**, far below ZoomEye (767) and ViCrop (502).
> 3. **Total Runtime:** Including initialization (~2.8s), BVS achieves **10.2s end-to-end latency**, still much faster than ZoomEye (23.9s), RAP (17.7s), and ViCrop (15.2s).
>
> **W3 & Q2: Acquisition optimization and scale parameterization.**
>
> **A3:** To optimize $\alpha_t(z)$ over the continuous spatial-scale domain, we adopt a hybrid strategy combining dense evaluation and local refinement:
>
> 1. **Optimization:** We first evaluate $\alpha_t(z)$ on a dense 3D grid (space × scale) to identify promising regions, then apply multi-start L-BFGS-B from top candidates, enabling accurate estimation of $z_{t+1} = \arg\max_{z \in X} \alpha_t(z)$.
> 2. **Efficiency:** With a small number of steps ($T \le 5$), the cost is dominated by GP updates. Evaluating $\alpha_t(z)$ over $N$ candidates costs $O(N t^2)$ at step $t$, which is negligible compared to MLLM forward passes.
>
> Regarding the noted inconsistency: $\ell(s)$ is correctly defined in Eq. (2) as **increasing with scale $s$**, meaning larger $\ell(s)$ yields smoother GP variation, consistent with the more gradual changes in coarse views. The reference to “small scale” is a typo, which we will correct.
>
> **Q3: Applicability of theoretical guarantees to the practical implementation.**
>
> **A4:** **Yes, the theoretical guarantees remain valid**, as our implementation preserves the key assumptions required by the GP-UCB framework.
>
> 1. **Kernel realization:** The scale-aware non-stationary kernel (Definition 2.1) follows the Gibbs formulation, and the arithmetic-mean approximation (Remark 2.3) is a standard realization (Paciorek & Schervish, 2006) that **preserves positive semi-definiteness (PSD)**. This ensures a valid GP prior on the $S \times \Omega$ manifold and does not violate the conditions for the sub-linear regret bound (Theorem 2.2). Empirically, it achieves **93.2% in 5 steps**, while a stationary kernel fails to match this even after 15 steps (Table 5).
>
> 2. **Oracle as noisy evaluator:** The prompt-based MLLM provides noisy observations $y_t = f(z_t) + \epsilon$. Although the noise may deviate from a Gaussian assumption, GP-UCB remains valid under **sub-Gaussian noise**, requiring only bounded uncertainty. The consistent gains across benchmarks (Tables 1–2) and models (InternVL, Qwen) indicate that this assumption holds in practice, enabling BVS to realize the predicted sub-linear regret behavior.

---

> > ### Author Rebuttal · Reviewer_XB8m · 2026-04-03
> >
> > The authors have clarified the previously missing details and provided additional experiments that further validate the strength of their approach. I maintain my initial score, since my original evaluation was already positive and the rebuttal has reinforced that assessment.

---

> > > ### Author Response · Authors · 2026-04-07
> > >
> > > **Response to Reviewer XB8m (Follow-up)**
> > >
> > > We sincerely appreciate your constructive engagement throughout the review process and for confirming that our clarifications resolved your reservations. Your rigorous examination of the "reasoning-aware" claims and efficiency metrics was instrumental in making our paper more robust. We are glad that the additional validation and the detailed end-to-end latency analysis were helpful in strengthening the final manuscript.

---

### Official Review · Reviewer_GLbc · 2026-03-11

**Soundness:** 3
**Presentation:** 1
**Significance:** 2
**Originality:** 3
**Overall Recommendation:** 5
**Confidence:** 3

**Summary:**

This paper proposes Bayesian Visual Search to find relevant regions of a high-resolution image input to an MLLM. The algorithm works by first using the attention rollout from the final query token back to image tokens, stopped at the middle layer of the network, to initialize the prior mean of a Gaussian Process. It then iteratively selects crops via GP-UCB - picking the region with the highest combination of estimated relevance and uncertainty - and queries an MLLM to get a 0-1 score on whether that crop contains the objects mentioned in the query. Each score is used to update the GP posterior. Finally, a 2D importance map is produced by evaluating the learned GP over a spatial grid and taking the max over zoom levels, then thresholding it to remove irrelevant pixel rows and columns from the original image. This content-aware squeezed image is fed into an MLLM to get the final answer.

**Compliance With Llm Reviewing Policy:**

Affirmed.

**Final Justification:**

The rebuttal has addressed my questions regarding some technical aspects of the paper. It has reinforced my initial positive assessment.

**Key Questions For Authors:**

- How is the final 2D importance map constructed from the GP?

- Is there existing work that explores seam carving for this task? Is it possible to use some sort of carving approach instead of removing rectangular regions of the image?

- Instead of the row-column squeezing, did the authors try a simple cropping approach using the BVS importance map?

**Limitations:**

The authors adequately highlight technical limitations. I do not foresee any negative societal impact.

**Strengths And Weaknesses:**

**Strengths**

- SOTA performance with fewer forward passes than most baselines. On HRBench-8K, BVS achieves 78% accuracy in under 5 forward passes vs 20-35 for ZoomEye.
- Elegant combination of UCB acquisition, zoom-aware kernel, and image consolidation. The squeezing step is reminiscent of seam carving and could be thought of as MLLM guided seam carving.
- The authors are transparent about where BVS doesn't help. Table 3 MMStar results showing performance drop on logical reasoning and math. This is understandable since those subsets are not about fine-grained visual detail.

**Weaknesses**
- Writing is poor. Key implementation details, like how zoom levels are discretized and how the final GP is evaluated to produce the 2D map are not clear. Even otherwise simple concepts, such as using the attention maps from middle layers, are masked in jargon and hard to parse.
- The regret proof is not a novel contribution. It basically shows the scale-aware kernel is valid and therefore the existing Srinivas 2009 GP-UCB bound applies.
- The oracle scoring prompt used during search explicitly ignores spatial relationships, scoring only noun presence. If the query hinges on a spatial relationship ("What is the girl on the left of the tree wearing?"), the oracle MLLM score might be suboptimal or even fail.

---

> ### Author Rebuttal · Authors · 2026-03-31
>
> **Response to Reviewer GLbc.**
>
> We sincerely appreciate your insightful and carefully considered feedback. We are encouraged that you recognize (1) the strong accuracy–efficiency trade-off of BVS, (2) the principled integration of GP-UCB, the zoom-aware kernel, and information consolidation, and (3) focusing on perception-oriented tasks. We also appreciate your implementation-focused questions, which help clarify key details. We respond below.
>
> **W1 & Q1: Writing clarity and key implementation details.**
>
> **A1:** We will further streamline the presentation and clarify the implementation. Below we directly address your questions:
> **1. How are zoom levels discretized?**
> We discretize the zoom levels as $S = {0.25, 0.5, \dots, 1}$ when computing the GP posterior mean. The base scale $s = 1.0$ corresponds to an observation window of $896 \times 896$ pixels to limit observation scale.
> **2. How is the final 2D importance map constructed from the GP?**
> We convert the GP posterior into the final input in three steps (Section 2.6):
> * **Step 1 (Aggregation):** Compute $\mu_T(s,x)$ and collapse scales via
>   $M(x) = \max_{s \in \mathcal{S}} \mu_T(s, x)$, yielding a 2D relevance map.
> * **Step 2 (Binarization):** Threshold with $\tau$ (default $0.2$):
>   $B(x) = \mathbf{1}[M(x) > \tau]$.
> * **Step 3 & 4 (Squeezing):** Extract minimal row/column sets $(\mathcal{R}, \mathcal{C})$ covering $B(x)$ and construct
>   $I_{sq} = I[\mathcal{R}, \mathcal{C}]$, preserving pixels whose row and column are selected.
>
> **W2: Novelty of the regret proof.**
>
> **A2:** We agree that our analysis builds on Srinivas et al. (2009); the contribution of Section 2.5 lies in establishing **formal guarantees for visual search in both convergence and efficiency**.
> Unlike heuristic methods (e.g., ZoomEye, ViCrop, RAP) without guarantees, BVS provides a principled foundation: the scale-aware non-stationary kernel is PSD and satisfies Mercer’s theorem on the $S \times \Omega$ manifold (Theorem 2.2), yielding a sub-linear regret bound.
> Furthermore, MIG analysis (Theorem 2.5) shows that BVS achieves the same convergence order as the optimal Bayesian strategy, making its efficiency theoretically grounded.
>
> **W3: The oracle scoring prompt ignores spatial relationships.**
>
> **A3:** We understand this concern. In contrast, this design is to stabilize the search process.
> Relation-heavy queries involve multiple objects (e.g., “the girl” and “the tree”), so a single region may not cover all entities. Enforcing strict spatial constraints at the scoring stage would penalize such partial observations for missing relational context, discarding useful regions early and decrease performance empirically.
>
> **Q2: BVS vs. Seam Carving.**
>
> **A4:** We appreciate you for this creative insight. To our knowledge, no existing work has explored seam carving for MLLM visual search. The related work (Appendix B) focuses primarily on crop-based policies. To directly address your suggestion, we conducted a comparison between our Row/Column Squeezing and a saliency-guided Seam Carving approach on the V* Benchmark using Qwen3-VL-8B, where the spatial importance map $M(x)$ (Section 2.6) was used as the pixel energy function.
>
> | Method | Retention Ratio | V* Accuracy (%) |
> | :--- | :---: | :---: |
> | **BVS (Squeezing)** | **0.12 (avg.)** | **93.2** |
> | Seam Carving | 0.90 | 83.3 |
> | Seam Carving | 0.50 | 82.7 |
> | Seam Carving | 0.30 | 84.3 |
> | Seam Carving | 0.10 | 84.8 |
> | Naive (No search) | 1.00 | 82.7 |
>
> **Findings and Analysis:**
> BVS achieves 93.2% accuracy while retaining only 12% of the image area, outperforming aggressive Seam Carving best by 8.4%.
> We analyse bad cases and argue this gap arises because Seam Carving introduces geometric distortion that disrupts spatial structure, whereas BVS’s subregion-preserving squeezing maintains object geometry and alignment for reliable reasoning.
>
>
> **Q3: Row-Column Squeezing vs. Simple Cropping.**
>
> **A5:** We evaluated a standard cropping policy following *ViCrop* using our BVS importance map. On the V* Benchmark with Qwen3-VL-8B, **squeezing (93.2%) significantly outperforms cropping (89.1%)**.
>
> As shown below, while both methods perform well on direct attributes, ViCrop shows consistent degradation across tasks:
>
> | Method              | Overall (%) | Direct Attributes (%) | Relative Position (%) |
> | :------------------ | :---------: | :-------------------: | :-------------------: |
> | ViCrop (Cropping)   |     89.1    |          92.2         |          84.2         |
> | **BVS (Squeezing)** |   **93.2**  |        **95.7**       |        **89.5**       |
> | **Gain**            |   **+4.1**  |        **+3.5**       |        **+5.3**       |
>
> We attribute this gap to a fundamental limitation of cropping: **Context–Resolution Inefficiency**.
> To cover spatially dispersed objects, cropping must expand and include irrelevant background, wasting tokens while reducing effective resolution on key entities.

---

> > ### Author Rebuttal · Reviewer_GLbc · 2026-04-01
> >
> > The authors have clarified missing details and conducted additional experiments that verify the strength of their approach. Although the writing remains a weakness, the authors can improve it in the camera-ready version. I maintain my initial score since it was based on the technical strength of the work.

---

> > > ### Author Response · Authors · 2026-04-07
> > >
> > > **Response to Reviewer GLbc (Follow-up)**
> > >
> > > Thank you for your follow-up and for the positive assessment of the strength of our work. We are especially grateful for your creative suggestion to compare BVS with seam carving, which significantly enriched our empirical analysis and highlighted the advantages of our squeezing strategy. We are committed to prioritizing the clarity of the writing and implementation details in the camera-ready version as you suggested.

---

### Official Review · Reviewer_oTGZ · 2026-03-12

**Soundness:** 3
**Presentation:** 3
**Significance:** 2
**Originality:** 2
**Overall Recommendation:** 3
**Confidence:** 3

**Summary:**

The paper proposes BVS, a visual search framework for improving fine-grained perception in ultra-high-resolution images with multimodal large language models. The method formulates region selection as a Bayesian optimization problem over spatial location and scale, guided by attention-based priors extracted from the MLLM. The framework iteratively samples image crops, evaluates them with the model, and aggregates informative regions to construct a compact visual representation for answering the query. Experiments on several fine-grained perception benchmarks show improved performance compared to existing search-based baselines.

**Compliance With Llm Reviewing Policy:**

Affirmed.

**Key Questions For Authors:**

1. See the weaknesses above.
2. BO is typically most suitable for relatively smooth functions, while crop relevance in visual perception tasks may be highly discontinuous. How does the method handle this issue in practice?

**Limitations:**

The paper includes a short impact statement but does not discuss technical limitations in detail.

**Strengths And Weaknesses:**

Strengths:

1. The paper addresses the problem of fine-grained perception in ultra-high-resolution images, which is an important challenge for multimodal large language models.

2. Empirically, the method achieves strong performance on several fine-grained perception benchmarks, outperforming existing search-based baselines.

Weaknesses:

1. The necessity of Bayesian Optimization is not fully justified, as the paper does not compare with other search strategies (e.g., greedy search or tree search).
2. The search process relies heavily on the attention-based prior obtained from the MLLM. If the prior fails to highlight the correct region, the subsequent BO search may be misled. A discussion or analysis of failure cases would strengthen the paper.
3. The paper evaluates efficiency mainly in terms of the number of forward passes. However, it would be helpful to report practical runtime or computational cost, especially since each search step requires an additional MLLM inference.

---

> ### Author Rebuttal · Authors · 2026-03-31
>
> **Response to Reviewer oTGZ.**
>
> We sincerely thank the reviewer for the constructive feedback and for recognizing both the significance of fine-grained perception in ultra-high-resolution (UHR) scenarios and BVS's strong empirical performance. We address your specific concerns in detail below.
>
> **W1: The necessity of Bayesian Optimization.**
>
> **A1:** We argue that the necessity of BO lies in (1) replacing rigid, discrete heuristic search mechanisms with a **continuous, uncertainty-aware, posterior-corrected search mechanism**; and (2) the ability to effectively combine prior and posterior search to simultaneously **improve both efficiency and performance**. As noted by Reviewers *bRSs* and *XB8m*, BVS offers a "principled, iterative, and self-correcting" optimization over a continuous spatial-scale manifold. Experiments in Section 3.2 and Table 1 compare it with alternative search methods:
> *   **Greedy Search:** *ViCrop* selects image crops based on fixed attention priors. Without uncertainty-driven exploration, it often gets stuck in local optima, resulting in lower accuracy (e.g., 69.2% on HRBench-8K).
> *   **Tree/Graph Search:** *ZoomEye* and *RAP* traverse discrete patches via tree-based search. While exploring broader regions, their discrete design and limited use of priors make them computationally expensive.
>
> By harnessing GP-UCB to balance prior exploitation with uncertainty-driven exploration, BVS addresses the efficiency of visual search in fine-grained perception. As Reviewer *GLbc* highlights, its strongest evidence is efficiency: on HRBench-8K, BVS achieves >78% accuracy in **<5 search steps**, whereas tree-based methods (ZoomEye/RAP) require **20 to 35 passes** for comparable or worse performance.
>
> **W2: The search may be misled if the prior fails.**
>
> **A2:** We fully understand this concern, however, mitigating the risk of a misleading prior is exactly one of the key advantages of BVS compared to a greedy, prior-only policy. The robustness of BVS is guaranteed by our prior-plus-posterior mechanism:
> 1. **Prior is merely an initialization:** The attention rollout only initializes the GP mean ($\mu_0$). It does not permanently constrain the search path.
> 2. **Search is driven by uncertainty:** The actual exploration relies on the GP-UCB acquisition function ($\alpha_t(z) = \mu_{t-1}(z) + \sqrt{\beta_t}\sigma_{t-1}(z)$).
> 3. **Dynamic self-correction:** If the prior completely misses the target, the unobserved regions will maintain high uncertainty ($\sigma$). As the search deepens, this uncertainty systematically inflates their acquisition weights, inevitably forcing the algorithm to explore these blind spots.
>
> **W3: Efficiency is reported mainly in forward passes rather than practical runtime.**
>
> **A3:** We agree that runtime and computational overhead are the most practical metrics for deployment. To address this, we provide a direct efficiency analysis from three perspectives: **forward passes**, **TFLOPs**, and **execution time** (see [figure](https://anonymous.4open.science/r/anonymous-90FC/vstarbench_three_panel_efficiency.png)) on the V\* benchmark with Qwen3-VL-8B, the BVS establishes a strictly dominant Pareto frontier across all three metrics: To reach its peak accuracy of 93.2% (at 5 search steps), BVS requires only **~165 TFLOPs** and **10.2s** end-to-end runtime (including the one-time prior initialization). In stark contrast, the strongest baseline ZoomEye (18 steps) achieves only 90.1% while demanding **767.5 TFLOPs** and **23.9s**. Thus, BVS is **2.3$\times$ faster** and consumes **5$\times$ less compute**.
>
> **Q1: BO is typically suited for smooth functions, whereas crop relevance can be discontinuous.**
>
> **A4:** To mitigate this issue, we exploit a practical property: **local continuity emerges when crops overlap at an appropriate scale, even if global behavior is discontinuous.** Our method is designed to preserve and leverage this scale-dependent local structure through two components:
>
> 1. **Matérn-based Foundation:** Our Scale-Aware Non-stationary Kernel builds on the Matérn function ($\nu=5/2$) rather than the Squared Exponential (RBF) kernel. Unlike RBF, the Matérn family does not assume infinite differentiability, allowing it to model functions with limited smoothness and abrupt local changes. This makes it better suited for capturing sharp semantic boundaries in crop relevance.
> 2. **Dynamic Scale-Awareness:** The validity of local correlation depends on scale. At coarse scales, small spatial shifts produce gradual changes, while at fine scales, the same shift can abruptly remove a target from the crop. To maintain usable local structure, our kernel dynamically enlarges the spatial length-scale during coarse-grained observations, effectively smoothing scale-induced discontinuities and stabilizing the BO process.

---

> > ### Author Rebuttal · Reviewer_oTGZ · 2026-04-04
> >
> > The newly provided runtime and TFLOP comparisons (W3) directly address my efficiency concern and are a valuable addition. The argument regarding GP-UCB's uncertainty-driven self-correction (W2) is conceptually sound, though an empirical demonstration of recovery under deliberately corrupted priors would be more convincing than the theoretical reasoning alone. For W1, the comparisons largely restate existing baselines from Table 1 rather than providing a controlled ablation, which would more directly isolate the contribution of Bayesian Optimization itself. I am willing to slightly adjust my assessment, but my reservations regarding ablation depth and the overall significance of the contribution remain.

---

> > > ### Author Response · Authors · 2026-04-07
> > >
> > > **Response to Reviewer oTGZ (Follow-up)**
> > >
> > > We sincerely thank the reviewer for the continued engagement and constructive feedback. To directly address your remaining reservations, we provide the controlled ablation to isolate the contribution of Bayesian Optimization, alongside an empirical demonstration of recovery under deliberately corrupted priors.
> > >
> > > **1. Empirical Demonstration of Recovery under Corrupted Priors**
> > > To provide an empirical demonstration of recovery under deliberately corrupted priors, we evaluate the search trajectories on the V* benchmark using the Qwen3-VL-8B backbone, where the ground truth initial attention prior is explicitly corrupted. Specifically, we apply a zero-mask to the attention values within a 1.5$\times$ expanded bounding box of the ground-truth target region to eliminate residual attention signals that might inadvertently guide the search. We then compare the step-by-step search trajectories of BVS (GP-UCB) against a Prior-Greedy baseline under this misleading initialization.
> > >
> > > The visualizations can be found here: [figure](https://anonymous.4open.science/r/anonymous-90FC/vstarbench_cases.png).
> > >
> > > **What the visualizations show:** In these cases, the corrupted prior initially misdirects the search to irrelevant background regions. The Prior-Greedy method becomes permanently trapped in these local optima, repeatedly sampling empty or incorrect regions. In contrast, BVS begins by exploring the false prior, but as those regions yield low posterior relevance, the uncertainty ($\sigma_t$) in the unexplored (but actually correct) regions systematically inflates. This mathematically forces the GP-UCB acquisition function to break away from the corrupted prior, explore the unobserved space, and locate the true target within the search budget. This provides direct empirical proof of the mechanism we theoretically outlined in our previous response.
> > >
> > > **2. Controlled Ablation Isolating Bayesian Optimization**
> > > To more directly isolate the contribution of Bayesian Optimization itself and provide further quantitative evidence, we conducted the controlled ablation on the V* benchmark. All evaluated methods share the exact same Qwen3-VL-8B backbone, the exact same attention prior initialization, the same 5-step search budget, and the same final answer pipeline. Only the search policy is changed:
> > >
> > > *   **(i) Prior-Greedy:** Greedily selects crops following the max value rank of the attention prior.
> > > *   **(ii) Posterior-Greedy:** Uses the same GP surrogate as BVS but selects crops by maximizing *only* the posterior mean ($\mu_t$), i.e., without uncertainty exploration ($\beta_t=0$).
> > > *   **(iii) Prior-Tree:** A prior-guided coarse-to-fine tree search with a 4-level hierarchy (896 $\rightarrow$ 672 $\rightarrow$ 448 $\rightarrow$ 224) and 4 children per node.
> > > *   **(iv) BVS (Ours):** Full GP-UCB optimization.
> > >
> > > | Method | Overall Acc. | Direct Attributes | Relative Position | Avg. Time / Sample |
> > > | :--- | :---: | :---: | :---: | :---: |
> > > | Prior-Greedy | 87.4% | 89.6% | 84.2% | 23.91s |
> > > | Posterior-Greedy| 88.5% | 92.2% | 82.9% | 19.34s |
> > > | Prior-Tree | 90.1% | 93.0% | 85.5% | 23.26s |
> > > | **BVS (Ours)** | **92.1%** | **93.9%** | **89.5%** | **22.92s** |
> > >
> > > This ablation isolates the contribution of the search strategy itself: even under the same prior and inference budget, replacing GP-UCB with either greedy or tree-based search consistently degrades performance. In particular, the 3.6% gap between Posterior-Greedy and BVS demonstrates that the gain does not come merely from maintaining a GP surrogate, but specifically from the **uncertainty-aware exploration**. These empirical results further validate the theoretical advantages of BO over existing search methods discussed in our previous response. Therefore, BO is not an interchangeable implementation detail; it is the core mechanism that effectively converts the prior into a robust and self-correcting search process.
> > >
> > > We will incorporate this controlled ablation and the visual demonstrations into the final version. We hope these targeted experiments resolve your remaining reservations.

---

### Official Review · Reviewer_bRSs · 2026-03-13

**Soundness:** 3
**Presentation:** 2
**Significance:** 3
**Originality:** 3
**Overall Recommendation:** 4
**Confidence:** 2

**Summary:**

The paper introduces BVS (Bayesian Visual Search), a framework designed to enhance fine-grained perception in Multimodal Large Language Models (MLLMs), particularly when handling ultra-high-resolution (UHR) images containing tiny or cluttered objects.
The authors aim to address a significant problem: current MLLMs often struggle with these tasks because they either lack efficient search strategies (prior-free) or rely on static heuristics that cannot be corrected during inference (prior-driven).
The proposed framework consists of three key components: Prior Attention Acquisition; Active Search via Scale-Aware BO; Information Consolidation.
The paper demonstrates that the proposed BVS significantly outperforms existing state-of-the-art search methods across fine-grained perception benchmarks.

**Compliance With Llm Reviewing Policy:**

Affirmed.

**Final Justification:**

Thanks for the authors' detailed responses, which successfully addressed my concerns on the latency trade-offs and the proxy model setup. I have no further questions. Since my original rating is already positive, I'd like to maintain it.

**Key Questions For Authors:**

Please refer to the weakness part.

**Limitations:**

yes

**Strengths And Weaknesses:**

Strength:
- Rather than relying on simple heuristics or greedy tiling, the paper provides a principled, iterative, and self-correcting search framework that treats perception as an optimization process.
- The approach directly addresses a critical bottleneck—efficiently processing high-resolution visual input—which is essential for deploying MLLMs in specialized fields (e.g., medical imaging, document analysis).
- The ablation studies effectively justify the necessity of the "early-stop" and "scale-aware" components.

Weakness:
- To my understanding, because BVS is an iterative optimization framework, it requires multiple inference steps. While the authors demonstrate superior accuracy, they should provide a more detailed analysis of the latency trade-offs, especially for real-time applications.
- The reliance on "proxy models" to generate priors for the base models is a potential point of failure. It is unclear how sensitive BVS is to the gap in reasoning capabilities between the proxy model and the base model.
- As shown in Table 3, BVS leads to a slight decline in performance on non-perceptual tasks like "Logical Reasoning". The authors should discuss whether the proposed BVS process might be losing broader contextual tokens that are vital for abstract reasoning.

---

> ### Author Rebuttal · Authors · 2026-03-31
>
> **Response to Reviewer bRSs.**
>
> Thank you for the thoughtful and constructive feedback. It is encouraging to see BVS recognized as a principled, iterative, and self-correcting framework, as well as for its strong performance on UHR benchmarks. We also appreciate your acknowledgment of efficient high-resolution processing and the supporting ablations on the early-stop and scale-aware components. Below, we address your concerns on efficiency, proxy priors, and non-perceptual tasks.
>
> **W1: Latency trade-off of iterative optimization.**
>
> **A1**: We agree that end-to-end latency is the key practical concern, and we provide a direct efficiency analysis from three perspectives: **forward passes**, **TFLOPs**, and **execution time** (see [figure](https://anonymous.4open.science/r/anonymous-90FC/vstarbench_three_panel_efficiency.png)). For forward passes, BVS reaches 93.2% accuracy with **5 search steps**, while ZoomEye requires 18 steps to reach 90.1%, corresponding to a 3.6× reduction in iterative steps. The compute gap is even larger: BVS requires **165 TFLOPs**, whereas ZoomEye requires 767 TFLOPs, and RAP and ViCrop require 447 TFLOPs and 502 TFLOPs, respectively. The same trend holds in latency, BVS takes about 7.4 s for the 5-step search and **10.2 s** end-to-end including the one-time prior initialization. By comparison, ZoomEye takes 23.9 s at 18 steps, RAP takes 17.69 s, and ViCrop takes 15.21 s. Therefore, BVS achieves the best accuracy with significantly lower latency than ZoomEye, while using far fewer search steps and much less computation.
>
> **W2: About the sensitivity to proxy models.**
>
> **A2**: We would like to clarify that the use of a unified proxy model Qwen2.5-VL served two purposes: (1) caching priors offline to  accelerate experimental iterations, and (2) demonstrating that BVS can achieve substantial performance gains even when initialized with priors from non-SOTA models. To directly address your concern and eliminate any potential misunderstandings, we conducted new experiments using the base models themselves to generate priors (i.e., without proxy). Specifically, Qwen3-VL-8B using its own self-generated prior achieves **90.5% accuracy** on the V\* benchmark in just 1 search step and gain  **92.5% accuracy** with 5 steps, highly competitive and still significantly outperforming existing baselines).
>
> **W3: About the slight drop on non-perceptual reasoning tasks.**
>
> **A3**: We appreciate the insightful observation and suggestion. To investigate whether BVS loses vital contextual tokens, we conducted  analysis on **the three abstract reasoning categories: Logi. Reason, Sci & Tech and Inst. Reason in MMStar**.
>
> (1) we were surprised to find that **BVS also provides benefits in these domains**. As shown below, it successfully corrects 20 previously unsolved samples by providing clearer visual evidence. This also leads to the overall drop being marginal (-17 out of 750 samples, or -2.2%).
>
>
> | Category     | Total | Naive $\checkmark$ / BVS $\times$ | Naive $\times$ / BVS $\checkmark$ | Net Change |
> | ------------ | ----- | --------------------------------- | --------------------------------- | ---------- |
> | Logi. Reason | 250   | 11                                | 6                                 | -5         |
> | Sci & Tech   | 250   | 17                                | 8                                 | -9         |
> | Inst. Reason | 250   | 9                                 | 6                                 | -3         |
> | **Total**    | 750   | 37                                | 20                                | -17        |
>
>
> (2) We manually examined the 37 degraded cases (Naive $\checkmark$ / BVS $\times$). We found that most errors actually result from the base model's zero-shot sensitivity to benign layout changes (such as cropping away irrelevant white borders). Only about **23.5% of these errors are genuinely caused by the loss of critical visual regions**. We also identified two types of tasks that tend to be especially sensitive to image squeezing operation, which may be important to avoid in future applications:
>
> - **Maps**: BVS may over-focus on the specifically queried entity, inadvertently excluding the surrounding geographic context (such as adjacent islands) that is essential for relative positioning.
> - **Line-connected Graphs (e.g., flowcharts, topologies)**: BVS usually finds the queried nodes effectively, but sometimes compresses the "empty" spaces that contain important connecting lines between nodes.

---

> > ### Author Rebuttal · Reviewer_bRSs · 2026-04-02
> >
> > Thanks for the authors' detailed responses, which successfully addressed my concerns. I have no further questions. Since my original rating is already positive, I'd like to maintain it.

---

> > > ### Author Response · Authors · 2026-04-07
> > >
> > > **Response to Reviewer bRSs (Follow-up)**
> > >
> > > We are very grateful for your final acknowledgment and for confirming that our responses fully addressed your concerns. Your insightful feedback on the latency trade-offs and the proxy model setup was particularly helpful in refining the practical positioning of BVS. We sincerely appreciate your support for our work!

---

### Decision · Program_Chairs · 2026-04-30

**Decision:**

Accept (regular)

**Comment:**

The paper proposes Bayesian Visual Search to enhance fine-grained perception in multimodal large language models for ultra-high-resolution images. The paper shows that the proposal outperforms existing state-of-the-art methods on fine-grained perception benchmarks needing fewer forward passes than most baselines.

Overall the reviewers found the proposal to be first principled and that the optimization problem of searching to be interesting, but there were still some concerns about the theoretical contributions on the proofs and additional experiments.  While the rebuttal addressed most of the issues, only one reviewer fully recommends the acceptance of the paper.

Reviewer bRSs has no further concerns, yet recommends a weak accept.

Reviewer oTGZ recommends a weak reject, and mentioned that the rebuttal addressed their concerns partially since the authors missed some empirical results which were provided in a further reply.  The reviewer didn't reply.

Reviewer GLbc recommended an accept and the rebuttal reinforced their initial positive assessment.

Reviewer XB8m recommended a weak accept the reviewer raised that there were no further concerns.

Given the strengths of the paper and the raised weaknesses, I recommend a weak accept.